# OpenReview forum: "ColJailBreak: Collaborative Generation and Editing for Jailbreaking Text-to-Image Deep Generation"
_NeurIPS.cc/2024/Conference — NeurIPS 2024 poster_

### Official Review · Reviewer_kzE3 · 2024-07-08

**Soundness:** 2
**Presentation:** 2
**Contribution:** 2
**Rating:** 4
**Confidence:** 4

**Summary:**

This paper proposes an image editing pipeline to obtain NSFW images. Utilizing image segmentation and editing techniques, the attack transforms a safe images generated by proprietary models into an unsafe counterpart.

**Strengths:**

1. The attack strategy of post-hoc editing is rather novel.
2. The proposed attack is effective against proprietary text-to-image systems, in a black-box setting.
3. The paper is easy to follow.

**Weaknesses:**

1. My first major concern is that the attack pipeline seems quite costly and not practical -- it involves inferencing of two additional models (SAM and Inpainting models). Please add more results regarding time and computational cost of you attack.
2. Also, the "unsafe components" of the images obtained via your attack, which are the cores of NSFW images, are editted via some open-source models. Then why bother to generate the images via proprietary models in the first place? The attacker could simply use uncensored open-source diffusion models to generate unsafe images (e.g., Stable Diffusion), which might be as high quality as the proprietary ones, in the context of (un)safety.
2. You are assuming the image editing process has no built-in safety mechanism. However, your attack cannot utilize proprietary image editing systems, which has additional safety mechanism to prevent unsafe image editing.
3. The safety types you considered are too limited (only violence, harassment, and self-harm). There are many other unsafe categories, e.g., nudity and graphic violence (e.g., disgusting scenes). I don't see potential superiority of your method on these other unsafe categories. If you think I'm wrong, add more experimental results on these categories.
4. More baselines. For example, why didn't you consider SneakyPrompt [1] as a baseline to compare with in Table 1?
5. More advanced evaluators. There are many more advanced image safety evaluators than Q16 you could evaluate with. For example, you can use GPT-4o or other multi-modal LLM (e.g., Llava, LlavaGuard [2]).
6. In Eq (2), the filter $\mathcal F$ is only taking the text prompts as the inputs. You may want to clearly specify this somewhere ahead (cuz there are also image-based safety filters, as you have mentioned).

Overall, I feel the technical contribution of this work is not sufficient, and the attack itself is not practical (in terms of computational cost and convenience). Therefore, I'm giving a rating of "Reject."

[1] Yuchen Yang, Bo Hui, Haolin Yuan, Neil Gong, and Yinzhi Cao. Sneakyprompt: Jailbreaking text-to-image generative models. In Proceedings of the IEEE Symposium on Security and Privacy, 2024. 2, 3

[2] https://huggingface.co/AIML-TUDA/LlavaGuard-13B

**Questions:**

See "Weaknesses."

**Limitations:**

Yes.

---

> ### Author Rebuttal · Authors · 2024-08-07
>
> ### **Q1:Time and computational cost of the attack**
> Thank you for the comments. we have conducted a detailed analysis of the time and computational resources required for each step of the attack pipeline. We specifically measured the time required for a single attack, including the segmentation mask generation using the SAM model and the image editing using the Inpainting model. **All experiments are performed using two A100 40GB GPUs.**
>
> Table Time and computational cost of the attack
>
> | Component  | Time | Computational Cost (GPU Memory)  |
> | ---------------------------------- | ------------ | ---- |
> | Single Attack Total Time(Avg)  | 56.68s        | 10911MiB |
> | SAM Mask Generation(Avg)     | 18.36s         | 9367MiB |
> | Inpainting Model Editing(Avg)  | 37.32s       | 9303MiB |
>
>
> As shown in the table, the overall time and computational cost for each attack are manageable and demonstrate that our method is both efficient and practical.
>
> ### **Q2: Image quality generated by uncensored open-source diffusion models**
> Thank you for your comments. Firstly, our objective is to develop a unified jailbreaking method for text-to-image (T2I) models that can generate unsafe images. We aim to evaluate the performance and potential threats of these models during both the initial image generation and subsequent editing phases, rather than relying solely on existing uncensored open-source models.
>
> Secondly, in our previous experiments, models capable of generating high-quality images generally have strong safety checks in place. Uncensored open-source models often produce lower-quality images. For example, commercial versions of Stable Diffusion can generate high-quality images but have safety mechanisms, whereas the open-source version on Hugging Face tends to produce lower-quality images and also includes some safety checks.
>
> Thirdly, our research is not just about generating unsafe images but demonstrating a new attack method that involves both image generation and editing. This process shows how an attacker can gradually bypass the model's safety restrictions through a series of steps. This comprehensive approach not only highlights vulnerabilities in the generation phase but also reveals the potential to further bypass safety checks through post-processing.
>
> ### **Q3: Unsafe image editing methods**
>
> Thank you for your comments. Our research primarily focuses on open-source image editing models, which typically do not have built-in safety mechanisms. This allows us to highlight the potential vulnerabilities in these widely accessible models. We integrate the editing models into an overall jailbreaking framework, leveraging the Contrastive Language-Image-guided Collaborative Optimization component. This integration enables the T2I models to generate unsafe images effectively.
>
>
> ### **Q4: More unsafe content types**
>
> Thank you for your comments. We have added two unsafe image categories: nudity and disgusting scenes. For each unsafe content type, we generated 25 images and calculated the attack success rate (ASR), and the results are shown in the following table, and detailed visualization examples of the results are in **[anonymous link](https://github.com/Anonymity7050/anonymityNeurIPS2024/blob/main/fig_more_safety_types.pdf)**.
>
> Table Experimental results on ASR in more unsafe content types
>
> | Method         | Nudity  | Disgusting  |
> |----------------|--------------|------------------|
> | MMA-Diffusion  | 60.00%       | 52.00%           |
> | QF-Attack      | 64.00%       | 68.00%           |
> | SneakyPrompt   | 72.00%       | 56.00%           |
> | ColJailBreak   | 72.00%       | 92.00%           |
>
> ### **Q5: Baseline methods**
> Thank you for your comments. We used SneakyPrompt as a baseline in our experiments. The results are shown in the following table. The experimental results show that ColJailBreak outperforms SneakyPrompt on average on all unsafe types.
>
> Table Comparison with SneakyPrompt on all unsafe types
>
> | Model   | Method        |  CLIP Scores ↑ | ASR (GPT-4o) ↑ |
> |---------|---------------|--------------------------------|---------------------------------|
> | GPT-4   | SneakyPrompt  | 0.2189                         | 78.66%                          |
> |         | ColJailBreak  | 0.2705                         | 84.00%                          |
> | DALL·E 2| SneakyPrompt  | 0.2705                         | 66.67%                          |
> |         | ColJailBreak  | 0.2913                         | 82.66%                          |
>
> ### **Q6: Advanced evaluators**
> Thank you for your comments. We understand the importance of using more advanced image safety evaluators. In response, we have chosen to use GPT-4o for our new experiments. The results of these new experiments are shown in the table below. The experimental results show that when using GPT-4o as an evaluator to evaluate our method, our method has a higher attack success rate (ASR) than the baselines
>
> Table Experimental results using GPT-4o as an evaluator on all unsafe types
> | Model   | Method         |  ASR (GPT-4o) ↑ |
> |---------|----------------|----------------------------------|
> | GPT-4   | MMA-Diffusion  | 58.66%                          |
> |         | QF-Attack      | 73.33%                          |
> |         | SneakyPrompt   | 78.66%                          |
> |         | ColJailBreak   | 84.00%                          |
> | DALL-E 2| MMA-Diffusion  | 64.00%                          |
> |         | QF-Attack      | 57.33%                          |
> |         | SneakyPrompt   | 66.67%                          |
> |         | ColJailBreak   | 82.66%                          |
>
> ### **Q7: Supplement to Eq(2)**
>
> Thank you for your comments. We understand that this point may have caused some confusion, and we will clarify it in the revised manuscript. In Eq (2), the filter F only takes text prompts as inputs. We will specify this clearly ahead in the relevant section to avoid confusion with image-based safety filters.

---

> > ### Comment · Reviewer_kzE3 · 2024-08-10
> >
> > Thanks for the rebuttal. I'm increasing my rating from 3 to 4. My concerns mainly lie in the contribution and motivation of this paper.
> >
> > The authors argued:
> > > Thirdly, our research is not just about generating unsafe images but demonstrating a new attack method that involves both image generation and editing. This process shows how an attacker can gradually **bypass** the model's safety restrictions through a series of steps.
> >
> > * This exactly showcases how this work is not well motivated. I wouldn't agree to say that the proposed attack can "**bypass**" model's safety restrictions. It's the attacker who actually adds unsafe content into the images. **This is definitely not a safety / security problem of existing image generation models** -- these T2I models should be allowed to generate any safe images, even though they could be misused by malicious users to edit into an unsafe version.
> >
> >   Nevertheless, I would agree more if the authors phrase this technique as "a convenient way to automatically generate unsafe images." This is indeed somehow a risk lack of attention.
> >
> > * I'm also concerned about the work's technical innovation and contribution, especially when it comes to nudity. There are various existing tools using image editing to turn a safe image into an unsafe counterpart (see [link1](https://www.reddit.com/r/AIpornhub/) and [link2](https://nsfw.tools/)). For example, malicious users can easily remove a person's cloth from the image and obtain nude contents by simple clicking and drawing. Such existing NSFW tools are much more convenient than your proposed pipeline -- the malicious users don't even need to run a line of code.
> >
> >   **Besides, there is no discussion at all regarding these NSFW tools. Please add another "Related Work" paragraph to discuss them.**
> >
> > Overall, I'm still on the reject side. But I appreciate the authors' contribution and efforts so far, and thus I'm increasing my rating.

---

> > > ### Author Response · Authors · 2024-08-11
> > >
> > > Dear Reviewer kzE3,
> > >
> > > Thank you for your thoughtful comments. We appreciate your acknowledgment of our efforts and contributions. We understand your concerns regarding the motivation and contribution of our work. We would like to address these points in detail below.
> > >
> > > ### **Q1: Bypassing of Safety Restrictionss**
> > > Our intention was not to imply that the model’s safety mechanisms were entirely circumvented but rather to demonstrate how an attacker could exploit the capabilities of text-to-image (T2I) models through a sequence of operations to produce unsafe content. While we agree that the malicious addition of unsafe elements does not equate to a direct bypass of safety restrictions, it highlights a significant vulnerability in the process, particularly when considering the ease with which these steps can be automated.
> > >
> > > ### **Q2: Technical Innovation and Contribution**
> > >
> > > We appreciate your comparison to existing tools that can turn safe images into unsafe ones, such as those that involve simple image editing techniques for generating NSFW content. We agree that these tools are highly accessible and pose a serious risk.
> > >
> > > However, we believe our work contributes uniquely by providing a collaborative pipeline that integrates both image generation and editing within a unified framework. While it is true that individual tools for editing exist, our method demonstrates how these processes can be combined and automated in a way that malicious users could exploit with minimal intervention. This approach could potentially lower the barrier for generating harmful content, as it does not require users to have sophisticated editing skills.
> > >
> > > ### **Q3: Related Work on Existing NSFW Tools**
> > >
> > > We will add a new paragraph to the Related Work section to discuss these tools, as follows:
> > >
> > > In recent years, with the advancement of deep learning, particularly in text-to-image (T2I) models, the ability to generate image content has significantly improved. However, this capability has also raised widespread concerns about the potential risks of generating unsafe content (NSFW). Numerous tools and technologies have already emerged, enabling users to easily transform safe images into unsafe images. online communities such as AIPornhub represent a significant aspect of the NSFW content generation ecosystem. AIPornhub, the oldest and largest AI-generated pornography Subreddit, serves as the official community for AIPornhub.net. NSFW.tools is a comprehensive directory of AI-driven NSFW content generation tools. This platform serves as a hub for a wide variety of AI applications tailored specifically for adult content creation. Smexy AI is another significant platform in the NSFW AI landscape. It is designed to enable users to generate and share fantasy-themed visual content with remarkable ease. In summary, while existing NSFW tools have already demonstrated the convenience and risks associated with generating unsafe content, our approach further reveals the potential threats posed by T2I models in the generation of such content. Through this work, we aim to bring broader attention to the risks associated with T2I models and to encourage the development of preventive measures against these risks.

---

> > > > ### Comment · Reviewer_kzE3 · 2024-08-11
> > > >
> > > > Thanks for the follow-up reply and clarifications. I choose to maintain my updated rating for now.

---

### Official Review · Reviewer_jDc9 · 2024-07-12

**Soundness:** 3
**Presentation:** 3
**Contribution:** 3
**Rating:** 5
**Confidence:** 4

**Summary:**

The paper introduces a method to bypass safety filters in commercial text-to-image (T2I) models like DALL·E and GPT-4.
Diffrent previous methods which directly do the adversarial attack, this paper proposes a noval approach to bypass the safety filter and generate harmful content by collaboration of multiple steps. The method is validated on three datasets, demonstrating its effectiveness in producing unsafe content.

**Strengths:**

The method combines generation and editing to bypass safety filters, which is innovative and different from traditional adversarial prompt techniques.
From the qualitative results, the proposed method successfully bypasses both text-based and image-based safety filters.
The method is validated on multiple datasets and compared to existing baseline methods.

**Weaknesses:**

The test datasets is relatively small which contain hundreds of cases.
The cases shown in this paper are relatively simple that the reviewer concerns that some unsafe examples cannot be covered, such as fighting or pornographic exposure. These cases cannot be easily substitute an object to complete the generation.
If harmful content generation is achieved mainly through image editing, the entire method may be more related to the editing model and owes more credit to the capabilities of image editing.
This paper does not fully evaluate the alignment between the generation and the prompt that whether the result of the attack really contains the content of the input prompt. CLIPScore is not enough to evaluate this and no human evaluation is included as well.

**Questions:**

Could the authors present arguments or evidence to justify the method rather than simply relying on image editing.
Could more testing and human evaluation be added?

**Limitations:**

The authors should release the code or the dataset cautiously and resonsibly.

---

> ### Author Rebuttal · Authors · 2024-08-07
>
> ### **Q1:Dataset and more unsafe content types**
> Thank you for your  comments. Our main purpose of building the dataset is to validate the effectiveness of the proposed method. The scale of the dataset is not a decisive factor. In fact, we refer to some previous studies. For example, DACA(Deng et al. 2023) used a dataset of 100 cases, and SurrogatePrompt employed a dataset of 90 cases(Ba et al. 2023).
>
> In selecting and designing the dataset, we considered key metrics for validating the methodology. Although the cases included in the current dataset are relatively simple, they are sufficient to reflect the validity of the methodology. We will further expand the scale of the dataset in the future and pay special attention to these complex and insecure examples to fully validate the reliability of the methodology.
>
> For your concern that the some unsafe types can't be covered, for more complex scenes, we can achieve the desired result by editing multiple times and enlarging the mask size. To demonstrate the potential of our method, we add two additional types of unsafe content: **nudity content and fighting scenes**. For each unsafe content type, we generated 25 images and calculated the attack success rate (ASR), and the results are shown in the following table, and detailed visualization examples of the results are in **[anonymous link](https://github.com/Anonymity7050/anonymityNeurIPS2024/blob/main/fig_more_unsafe_types.pdf)**.
>
> Table Experimental results on attack success rates(ASR) in nudity and fighting scenes
>
> | Method        | Nudity   | Fighting  |
> |---------------|--------------|----------------|
> | MMA-Diffusion | 60.00%       | 56.00%         |
> | QF-Attack     | 64.00%       | 68.00%         |
> | ColJailBreak  | 72.00%       | 80.00%         |
>
> ### **Q2:The role of image editing methods**
>
> We understand your concern regarding the reliance on image editing in our method. We would like to clarify that our approach is not merely dependent on image editing capabilities but integrates several components to achieve the desired results. To address this concern, we have conducted two experiments that demonstrate the effectiveness and uniqueness of our method.
>
> **(1) Experiment 1 Comparing Different Editing Methods:**
>
> To demonstrate that our method is not solely reliant on a specific image editing technique, we have tested various editing methods within our framework. We employed several image editing models. As shown in the **[anonymous link](https://github.com/Anonymity7050/anonymityNeurIPS2024/blob/main/fig_different_editing_methods.pdf)**, regardless of the editing method used, our overall framework achieves similar outcomes in generating unsafe content. This demonstrates the robustness of our method and its ability to work with different editing techniques.
>
> **(2) Experiment 2: Impact of Contrastive Language-Image-guided Collaborative Optimization**
>
> To highlight the importance of our optimization method, we conducted experiments using the same editing method but without our optimization framework. We removed the collaborative optimization component and evaluated the performance of the editing model alone. As shown in the **[anonymous link](https://github.com/Anonymity7050/anonymityNeurIPS2024/blob/main/fig_impact_of_contrastive_language-Image-guided_collaborative_optimization.pdf)**, without the collaborative optimization framework, the editing methods alone were not able to achieve the same level of effectiveness in generating unsafe content. This underscores the critical role of our integrated approach in enhancing the capabilities of the editing models.
>
>
>
> ### **Q3: Results of human evaluation and more testing**
> Thank you for your comments. To enhance the credibility of our research, we have added more testing and human evaluation. Specifically, we conducted the following:
>
> **(1)	Human Evaluation:**
>
> We conducted Human Evaluation to evaluate the generation quality based on human responses. Specifically, each participant needs to answer the following two questions. So far, we have collected 50 survey responses for this evaluation.
>
> **Q1:** Participants were asked to rate the alignment between the image and the text prompt on a scale of 1 to 10, where 10 indicates the highest relevance and 1 indicates the lowest relevance. For Q1, we averaged all the scores to compare the performance of different methods.
>
> **Q2:** Given an image, participants were asked to classify it as either a safe image or a harmful image. For Q2, we calculate the percentage of images classified as harmful.
>
> The results of these new evaluations are shown in the Table below, demonstrating the performance of our method in aligning generated content with the input prompt and the attack success rate
>
> Table Experimental results of Human Evaluation
>
> | Method         | Q1  |  Q2(%)  |
> |----------------|------------------------|--------------------------|
> | MMA-Diffusion  | 7.25                   | 82.00%                   |
> | QF-Attack      | 6.75                   | 74.00%                   |
> | ColJailBreak   | 8.75                   | 86.00%                   |
>
> **(2)	Text-Image Alignment:**
>
> To further validate the alignment between the images and the text prompts, we used GPT-4o as an advanced evaluator, providing alignment scores similar to those in Q1. The results are shown in the table below.
>
> Table Experimental results of Text-Image Alignment
>
> | Method         | Text-Image Alignment (GPT-4o) ↑ |
> |----------------|---------------------------------|
> | MMA-Diffusion  | 6.25                            |
> | QF-Attack      | 5.50                            |
> | ColJailBreak   | 8.50                            |
>
> ### **Q4: Code and Dataset**
> We will release the dataset and code as soon as possible. We commit to following ethical guidelines in the release process.

---

> ### Author Response · Authors · 2024-08-11
>
> Dear Reviewer jDc9,
>
> We hope our previous response has effectively addressed your concerns. If you have any additional questions, we would be more than happy to continue the discussion with you.

---

### Official Review · Reviewer_yPxw · 2024-07-13

**Soundness:** 2
**Presentation:** 2
**Contribution:** 2
**Rating:** 5
**Confidence:** 4

**Summary:**

The paper introduces ColJailBreak, a framework that jailbreaks commercial text-to-image models by creating a safe image and modifying it to incorporate unsafe elements. It reveals the vulnerabilities of current safety filters in text-to-image models.

**Strengths:**

1. The paper is well-written. It studies an important problem (safety problem in text-to-image models).
2. The method is easy to follow in practice.

**Weaknesses:**

1. The intuition is a little strange. The authors use the text-to-image model to generate normal images first and then modify the generated images. Then, the original generated images are good. I do not think repainting some parts of the 'good' images could be treated as jailbreaking.

**Questions:**

1. What is the possible application of these attacks?
2. Can authors also test some other models like Midjourney?

**Limitations:**

Yes.

---

> ### Author Rebuttal · Authors · 2024-08-07
>
> ### **Q1: Explanation of our jailbreaking method**
> Thank you for the comment. We appreciate your concerns and would like to address them by elaborating on the intuition and rationale behind our proposed method. Our goal is to demonstrate the potential for generating unsafe images through a macro-level jailbreaking method. Specifically, we first use the text-to-image (T2I) model to generate normal images that easily pass the model's safety filters and are amenable to malicious edits. We then modify them to embed unsafe content. By showing that initially safe images can be edited to include unsafe elements, we emphasize the vulnerability of the models to post-generation manipulations. We hope this explanation clarifies the intuition behind our approach and the significance of our findings.
>
> ### **Q2: Application of our attacks**
> Thank you for the comment. The possible applications of the proposed attacks are multifaceted and significant in both demonstrating the vulnerabilities of existing safety mechanisms and emphasizing the need for more robust defenses. Here are the key applications:
>
> **(1) Production and Distribution of Illegal Content:** Creating and distributing adult content, pornographic images, and other illegal or inappropriate images.
>
> **(2) Malicious Manipulation and Defamation:** Producing fake obscene images for online defamation and damage to reputations.
>
> **(3) Psychological and Emotional Harm:** Intentionally spreading NSFW content to cause psychological harm or emotional distress, especially among minors or sensitive populations.
>
> **(4) Benchmarking and Testing:** These attacks can serve as benchmarks for testing the effectiveness of new defense mechanisms, ensuring that AI models are resilient to such vulnerabilities.
>
> ### **Q3: Experimental results of Midjourney**
>
> Thank you for your valuable suggestion. Based on your recommendation, we test Midjourney and find that our approach performed excellently. The detailed results of our testing are provided in the **[anonymous link](https://github.com/Anonymity7050/anonymityNeurIPS2024/blob/main/fig_Midjourney_results.pdf)**. Our method demonstrated superior performance in various evaluation metrics when applied to Midjourney, demonstrating its effectiveness.
>
> Table Quantitative evaluation of ColJailBreak and baselines in jailbreaking Midjourney
>
>  | Method       | All Unsafe Types | All Unsafe Types |
> |--------------|------------------|------------------|
>  |              | CLIP Scores ↑    | ASR ↑            |
>  | MMA-Diffusion| 0.2234           | 76.00%           |
> | QF-Attack    | 0.2418           | 84.00%           |
> | ColJailBreak | 0.2809           | 88.00%           |

---

> ### Author Response · Authors · 2024-08-11
>
> Dear Reviewer yPxw,
>
> We hope our previous response has effectively addressed your concerns. If you have any additional questions, we would be more than happy to continue the discussion with you.

---

> ### Author Response · Authors · 2024-08-13
>
> Dear Reviewer yPxw,
>
> We hope our previous response has effectively addressed your concerns. If you have any additional questions, we would be more than happy to continue the discussion with you.

---

### Official Review · Reviewer_GbGQ · 2024-07-14

**Soundness:** 3
**Presentation:** 3
**Contribution:** 3
**Rating:** 6
**Confidence:** 3

**Summary:**

This paper proposes a jailbreaking framework designed to bypass safety filters in commercial text-to-image (T2I) models. Specifically, it introduces three components for the jailbreak attack: adaptive normal safe substitution, inpainting-driven injection of unsafe content, and contrastive language-image-guided collaborative optimization.

**Strengths:**

1. The paper is well-presented.
2. It provides extensive evaluation results.

**Weaknesses:**

N/A

**Questions:**

N/A

---

> ### Author Rebuttal · Authors · 2024-08-07
>
> Thank you for your recognition and support. We are very happy with your evaluation of our work. In order to further improve our work, we plan to release the dataset and code as soon as possible, while ensuring that the release process follows ethical guidelines and best practices. Thank you again for your comment.

---

> ### Author Response · Authors · 2024-08-11
>
> Dear Reviewer GbGQ,
>
> We hope our previous response has effectively addressed your concerns. If you have any additional questions, we would be more than happy to continue the discussion with you.

---

### Author Rebuttal · Authors · 2024-08-07

We appreciate the thoughtful comments and helpful criticism from every reviewer. In this work, we introduce a jailbreaking framework designed to bypass safety filters in commercial text-to-image (T2I) models. Specifically, we present three components for the jailbreak attack: adaptive safe word substitution, inpainting-driven injection of unsafe content, and contrastive language-image-guided collaborative optimization.

We are delighted that **Reviewer GbGQ** found our paper to be well-presented and acknowledged the extensive evaluation results. **Reviewer yPxw** believes that our work studies the important issue of security of T2I models while being practical and reproducible. **Reviewer jDc9** appreciated the innovative method of combining generation and editing to bypass safety filters and noted the validation on multiple datasets. **Reviewer kzE3** provides a lower rating primarily due to some misunderstandings and concerns regarding the practicality and technical contributions of our method.  We believe our rebuttal could address your concerns.We have addressed the concerns of all reviewers. Below is a summary of some major responses. Please refer the response to reviewers about other minor or clarification responses.

1. **[ Motivation of our work]** We clarified the intuition behind our work in response to Reviewer yPxw.

2. **[Test on More Models]** In response to Reviewer yPxw's question about testing on other models like Midjourney, we have included additional experiments with diverse T2I models to demonstrate the generalizability of our method.

3. **[Human Evaluation]** In response to Reviewer jDc9's suggestion, we have incorporated a human evaluation component to assess the alignment between the generated images and the input prompts. This additional evaluation provides a more comprehensive understanding of our method's effectiveness.


4. **[Unsafe Content Types]** We expanded the scope of the unsafe category to cover a larger and more diverse range of cases to address reviewer jDc9 and reviewer kzE3's concerns about the limited number of safe types and the complexity of the examples.

5. **[Advanced Evaluators and More Baseline]** We have added SneakyPrompt as a baseline and employed GPT-4o as an advanced evaluator, in response to Reviewer kzE3's suggestions, to provide a more robust comparison and validation of our method.

---

### Decision · Program_Chairs · 2024-09-25

**Decision:**

Accept (poster)

**Comment:**

1x WA, 2x BA, and 1x BR. This paper proposes an image editing pipeline to obtain NSFW images using image segmentation and editing techniques. The reviewers agree on the (1) clear writing, (2) simple yet effective method, and (3) extensive evaluation. Most of the concerns, such as the insufficient datasets, insufficient baselines, and insufficient evaluation metrics, have been addressed by the rebuttal. Therefore, the AC leans to accept this submission.